# Core-Dependent Desorption Behavior of Polyelectrolyte Microcapsules in NaCl and Na_2_SO_4_ Solutions

**DOI:** 10.3390/polym17121706

**Published:** 2025-06-19

**Authors:** Alexey V. Dubrovskii, Aleksandr L. Kim, Sergey A. Tikhonenko

**Affiliations:** 1Institute of Theoretical and Experimental Biophysics Russian Academy of Science, Institutskaya St., 3, 142290 Puschino, Russia; dav198@mail.ru; 2Moscow Polytechnic University (Moscow Polytech), Bolshaya Semyonovskaya St., 38, 107023 Moscow, Russia

**Keywords:** polyelectrolyte microcapsules, PMC, polystyrene, CaCO_3_, MnCO_3_

## Abstract

Polyelectrolyte microcapsules (PMCs) have a wide range of applications in fields such as medicine, pharmacology, diagnostics, etc., and can be used as targeted drug delivery vehicles, diagnostic systems and smart materials. However, the existing research indicates that the type of core can influence the properties of the PMC shell. Consequently, we hypothesized that the type of core used for the formation of the PMC may also affect the desorption of the shell’s polyelectrolytes. In this study, the desorption of polyelectrolytes of PMCs, formed on polystyrene cores (PMC_Ps_) and MnCO_3_ (PMC_Mn_) and CaCO_3_ cores (PMC_Ca_), incubated in either NaCl or Na_2_SO_4_ solution, was investigated. It was demonstrated that the low ionic strength of the solution (up to 200 mM NaCl) has a negligible effect on the desorption of PMC_Ca_. However, in the case of PMC_Ps_ and PMC_Mn_, an increase in desorption was observed at 100 and 200 mM NaCl. Increasing the ionic strength to 1000 mM and 2000 mM resulted in a gradual increase in the desorption of the polyelectrolytes PMC_Ca_ and PMC_Mn_, while for PMC_Ps_, the maximum desorption was already observed at 1000 mM. Additionally, an increase in desorption was detected upon incubation in various concentrations of sodium sulfate (5–50 mM), although the desorption did not differ significantly across all types of PMCs. Nevertheless, for PMC_Mn_, the maximum desorption was observed at a sodium sulfate concentration of 50 mM, whereas for other types of capsules, the maximum desorption occurred at a concentration of 100 mM. These results support the hypothesis that the type of core used in the formation of PMCs influences the desorption of the shell polyelectrolyte.

## 1. Introduction

The surface modification method based on polyelectrolyte multilayer (PEM) deposition was invented in the 1990s by Dekker and Hong [1]. It is considered to be one of the most efficient methods capable of modifying a surface regardless of its shape. In addition, the layer-by-layer (LbL) polyelectrolyte multilayer assembly method is a versatile way to modify the surface and create new nanostructures. It is based on the alternate deposition of polycation and polyanion solutions on the substrate, usually with washing steps in between. This technology has found many applications in leading fields of science: the controlled release of bioactive compounds [2,3], tissue engineering [4,5], the creation of biosensors [6,7], environmental remediation and the retention of pollutants [8,9].

Surfaces coated with polyelectrolyte multilayers are effective colloidal carriers. The physicochemical properties of different polyelectrolytes allow the formation of different types of chemical bonds (electrostatic bonding, hydrogen bonding, π-π stacking, metal–ligand coordination, ionic and hydrophobic interactions) between the polyelectrolyte and organic or inorganic agents. As a result, polyelectrolyte multilayers can be applied to a wide range of macromolecules, from synthetic polyelectrolytes and biomacromolecules to nanoparticles of various metals [10]. Thus, this technology can be used to load/release bioactive compounds such as drugs [11], proteins [12], enzymes [13,14], nucleic acids [15,16], amino acids [17] and others, as well as contaminants such as dyes [18], pesticides [19,20], humic acids [21,22], heavy metal ions [23,24] and others [25]. In addition, polyelectrolyte multilayers can impart antimicrobial properties to flat surfaces or be used as substrates in tissue engineering using natural polyelectrolytes.

In the context of the polyelectrolyte multilayers discussed above, it should be noting that numerous parameters such as pH, ionic strength, charge density, etc. [26,27,28,29,30,31], affect the LbL process and the resulting PEMs. For instance, the structure and density of interpolyelectrolyte complexes or multilayers can depend significantly on the concentration of low-molecular-weight salt ions. These compounds can shield the charges of polyelectrolytes, leading to the desorption of macromolecules (polyelectrolytes) [32]. However, the conditions for desorption (e.g., the salt concentration) also depend on the symmetry of the structure, specifically the symmetry of the electrostatic field that maintains the opposite charges in the bound (“condensed”) state. For example, according to theoretical calculations, the adsorption of a linear chain on an oppositely charged planar surface [33] occurs at a critical surface charge density (ch.d.)_plan_~r_D_^−3^ (rD is the Debye shielding length). However, changing the surface geometry from planar to cylindrical [34] and spherical [35] requires a higher surface charge density for the chain binding of (ch.d.)_cyl_~r_D_^−2^ and (ch.d.)_sph_~r_D_^−1^. Therefore, the desorption of polyelectrolyte multilayers formed on spherical particles may show a different behavior. An example of such spherical systems is polyelectrolyte microcapsules.

Polyelectrolyte microcapsules (PMCs) are spherical microcontainers produced by LBL technology [36]. They are produced by the alternating adsorption of positively and negatively charged polyelectrolytes onto dispersed nano- and microsized particles [37,38,39,40]. PMCs were first obtained in 1998 and have since been the subject of active research in the field of polymer nanotechnology [40,41].

Polyelectrolyte microcapsules have a wide range of applications in various fields such as medicine, pharmacology, diagnostics, etc. Microcapsule shells can be made from polyelectrolytes with different properties, allowing them to be optimized for specific applications [42,43]. Polyelectrolytes can be divided into biodegradable and non-biodegradable, depending on whether enzymes are able to degrade the polyelectrolyte in question. Non-biodegradable polyelectrolytes, such as polystyrene sulphonate and polyallylamine, are resistant to biodegradation by natural microbial and enzymatic processes. Polyelectrolyte microcapsules made from these materials are able to withstand harsher incubation conditions for longer periods of time. Due to this property, PMCs can be used in activities where it is necessary to provide increased material resistance to external factors, for example in industrial and medical diagnostics to determine the pH of the environment [44], the concentration of low-molecular-weight compounds [45,46,47,48,49,50] and the surface charge of metals [51]; in industrial or domestic water treatment for wastewater treatment by sorption [23]; and other activities. Biodegradable polyelectrolytes, such as polyarginine and polylysine, are susceptible to degradation by proteolytic enzymes, resulting in the release of encapsulated substances into the incubation medium. Polyelectrolyte microcapsules containing these polymers can be used in various applications such as targeted drug delivery [44,52,53], prolonged action and controlled release [54,55,56,57]. However, it is known that the structure of the polyelectrolyte, its weight and charge density and the conditions of the incubation medium can affect the shell properties of PMCs and the desorption behavior of their polyelectrolytes [58,59,60]. Therefore, studies are carried out to investigate the influence of incubation medium conditions on the desorption behavior of the polyelectrolytes of microcapsules.

In our previous work (Dubrovskii et al. [61]) the destruction of microcapsule shells (PAH/PSS)_3_ containing protein was studied by fluorescence microscopy. PMCs were incubated in a medium with different concentrations of NaCl and (NH_4_)_2_SO_4_, after which the supernatant was collected. The amount of dissociated FITC-PAH was determined by fluorescence intensity. It was shown that the desorption of the polyelectrolyte of the microcapsules increased with the increasing ionic strength of the solution, with the highest desorption observed in a 2M NaCl solution. The authors explained this effect by the loosening of the PMC shell under the influence of ionic strength.

Polyelectrolyte desorption from microcapsules formed on protein-free CaCO_3_ particles was also investigated in our next work (Musin et al. [62]). Similar to the previous study, it was shown that increasing the ionic strength of the solution leads to an increase in the desorption of the polyelectrolytes of microcapsules (PMCs). However, in this study it was shown that increasing the temperature increased polyelectrolyte desorption in microcapsules formed on protein-free CaCO_3_ particles, whereas the opposite effect was shown in the work of Dubrovsky et al. as desorption decreased [61]. This may be due to the different internal structure of polyelectrolyte microcapsules formed on protein-free and protein-containing CaCO_3_ particles. Microcapsules formed on protein-free CaCO_3_ particles have a complex internal interpolyelectrolyte structure, whereas microcapsules formed on protein-containing CaCO_3_ particles have a pronounced shell [63].

Based on the aforementioned studies, it can be hypothesized that polyelectrolyte microcapsules formed on different types of cores may exhibit varying desorption behaviors of the shell polyelectrolytes. Previously, we demonstrated that the buffering capacity of polyelectrolyte microcapsules formed on polystyrene (PS) cores and CaCO_3_ cores differed significantly [64]. Given that the buffering capacity of microcapsules is attributed to the free amino groups of polyallylamine, it is reasonable to assume that the desorption of this polyelectrolyte will also vary depending on the type of core used. This assumption is further supported by the fact that, according to a literature analysis [65], the physicochemical properties and morphology of polyelectrolyte microcapsules formed on CaCO_3_ cores (PMC_Ca_), polystyrene particles (PMC_Ps_), and MnCO_3_ cores (PMC_Mn_) differ significantly. Specifically, microcapsules formed on PS and MnCO_3_ cores have shell thicknesses ranging from 10 to 60 nm. However, the shell thickness of PMC_Mn_ and PMC_Ps_ changes in a different way depending on the number of layers: PMC_Mn_ shells do not thicken, whereas PMC_Ps_ shells increase in thickness with additional layers [65]. In contrast, microcapsules formed on CaCO_3_ cores lack a distinct shell and instead consist of a spherical polyelectrolyte complex with an intricate channel-like structure [36]. Furthermore, changes in pH and ionic strength affect the size of PMC_Ca_, PMC_Mn_, and PMC_Ps_ differently: PMC_Ps_ expand in alkaline conditions and shrink at a low pH [66], while PMC_Ca_ remain unchanged in size across varying pH levels [67,68]. Increasing ionic strength reduces the size of PMC_Ps_ [26], whereas PMC_Ca_ remain unaffected [69]. In alkaline environments, PMC_Ps_ and PMC_Mn_ exhibit different size behaviors: PMC_Ps_ with 10 and 12 layers shrink by nearly half (from pH 11.5), while those with 14 and 16 layers double in size [65]. In contrast, PMC_Mn_ increase in size (by threefold) regardless of the number of shell layers [65]. Additionally, PMC_Ps_ demonstrate greater stability under highly alkaline conditions, enduring pH 12.5 for 30 min [26], whereas PMC_Ca_ and PMC_Mn_ degrade immediately [26,66]. This observation highlights the differences in the stability of polyelectrolyte layers among these capsule types and underscores the need to investigate the desorption of shell polyelectrolytes in microcapsules formed on PS and MnCO_3_ cores.

Therefore, the aim of this work is to study the desorption of polyelectrolytes of PMCs ((PAH/PSS)_3_PAH composition) formed on different types of cores: polystyrene cores and CaCO_3_ and MnCO_3_ cores.

## 2. Materials and Methods

Polystyrenesulfonate sodium (PSS) and polyallylamine hydrochloride (PAH) with a molecular mass of 70 kDa Sigma (Merck KGaA, Darmstadt, Germany), fluorescein isothiocyanate (FITC) Sigma (Merck KGaA, Darmstadt, Germany); ethylenediaminetetraacetic acid (EDTA), dimethylformamide (DMFA), calcium chloride (CaCl_2_ × 2H_2_O), sodium chloride, sodium sulfate and sodium carbonate from Reahim (Reahim AO, St. Petersburg, Russian) were used. Polystyrene particles measuring 5 microns with -COOH groups were purchased from Polymer Latex (Saint Petersburg, Russia).

### 2.1. Preparation of Fluorescently Labeled PAH

The polyelectrolyte (PAH) was labeled with FITC using a procedure adapted from our previous work [70]. FITC’s conjugation to PAH was carried out in borate buffer (50 mM, pH 9.0). To a stirred solution (300–400 rpm) of the polyelectrolyte (10 mg/mL), FITC was introduced slowly at a molar ratio of 1:100 (FITC:PAH) [61]. This labeling reaction, conducted for 1.5–2 h at room temperature, was followed by overnight dialysis against a large volume of water (10 L) for the purification of the FITC-PAH conjugate.

### 2.2. Preparation of CaCO_3_ and MnCO_3_ Microspherulites

The synthesis involved coprecipitation [61,71]. Under constant stirring, a 0.33 M solution of CaCl_2_ (or MnCl_2_) was introduced into an equal concentration (0.33 M) of Na_2_CO_3_. This rapid mixing phase lasted 30 s. Subsequently, the suspension was held static to enable complete particle precipitation and a controlled “ripening” process, tracked using light microscopy. Following established isolation procedures [70], the supernatant was decanted, and the precipitate underwent water washing prior to PMC preparation. Microparticle size analysis yielded an average diameter of 4.5 ± 1 (CaCO_3_) and 3 ± 1 μm (MnCO_3_) matching our prior synthesis.

### 2.3. Preparation of Polyelectrolyte Microcapsules Formed on CaCO_3_ and MnCO_3_

The synthesis of polyelectrolyte microcapsules (PMCs) via layer-by-layer (LbL) adsorption onto sacrificial carbonate cores followed our previously detailed procedure [72]. Microspherulites (CaCO_3_ or MnCO_3_) served as templates for the alternating deposition of PAH and PSS from aqueous solutions (2 mg/mL polymer + 0.5 M NaCl). To ensure surface cleanliness after each adsorption, the templates underwent three washes with 0.5 M NaCl solution, utilizing centrifugation for separation to eliminate non-adsorbed polymer chains. After depositing the desired number of bilayers, the core was dissolved by incubation in 0.2 M EDTA for 12 h. This dissolution step initiates the formation of an interpolyelectrolyte complex within the nascent capsule interior [72]. The final purification involved triple rinsing with water to remove core remnants. PMC size distribution and zeta potential analyzed by dynamic light scattering (Malvern Zetasizer Nano ZS, London, UK) yielded average diameters of 4.5 ± 1 μm (CaCO_3_-derived) and 3 ± 1 μm (MnCO_3_-derived), replicating the dimensions reported in our earlier study.

### 2.4. Preparation of Polyelectrolyte Microcapsules Formed on Polysteryne Particles

PMCs with polystyrene templates were prepared using a modified layer-by-layer (LbL) protocol adapted from our carbonate-core methodology [64,72]. The sequential adsorption of PAH and PSS (2 mg/mL each in 0.5 M NaCl) onto PS microparticles was performed. Following each deposition cycle, unbound polymer was removed through triple centrifugation washes in 0.5 M NaCl. After achieving the target bilayer count, the cores were dissolved in DMF (12 h). The resultant capsules underwent three aqueous washes to eliminate template residues. Morphological homogeneity was verified by light microscopy, revealing microcapsules with a 5 ± 0.2 μm average diameter and 22.6% polydispersity index.

### 2.5. Registration of FITC-Labeled PAH Desorption from Polyelectrolyte Capsules

The fluorescence spectroscopy analysis of microcapsule dissociation followed our established methodology [70], employing FITC-labeled PAH (λ_ex_ = 525 nm) in one capsule layer. Samples were centrifuged (3000 rpm, 1 min), after which 10 μL supernatant was diluted 40-fold to optimize the detection range. Fluorescence intensity measurements used a Cary Eclipse spectrofluorometer (Agilent, Santa Clara, CA, USA) with a 1 cm thermostatted cuvette, exciting the samples at 273 nm. Between measurements, the samples were vortexed and returned for continued incubation. Spectra acquisition monitored dissociation kinetics via FITC signal intensity.

### 2.6. Statistical Data Analysis

For each measurement of fluorescence intensity, the mean values and relative standard deviations were calculated. The number of replicates (N) was 6. The significance of the differences was assessed using an independent two-sample *t*-test (Student’s *t*-test), with *p* ≤ 0.05 being considered significant.

## 3. Results and Discussion

To investigate the desorption of polyelectrolytes from microcapsules formed on CaCO_3_ cores (PMC_Ca_), polystyrene particles (PMC_Ps_), or MnCO_3_ cores (PMC_Mn_), microcapsules with the composition (PAH/PSS)_3_/PAH were used, where the outermost PAH layer was fluorescently labeled. For this purpose, polyelectrolyte microcapsules were prepared through layer-by-layer (LbL) adsorption of the polyelectrolytes polystyrene sulfonate (PSS) and polyallylamine (PAH) onto CaCO_3_, MnCO_3_ or PS particles, followed by core dissolution. A schematic representation of the preparation process for polyelectrolyte microcapsules is provided in Figure 1.

To determine the desorption of the outer polyelectrolyte layer of PMCs, FITC-labeled polyallylamine (PAH) was used to form the outer positively charged layer of the PMC shell, and PAH without a fluorescent label was used to form the other positively charged layers. The polyelectrolyte microcapsules were then incubated in a solution containing a certain concentration of NaCl or Na_2_SO_4_, and the fluorescence intensity of the supernatant was measured.

In our previous work [62], the effect of 0.2M and 2M NaCl solutions on the desorption of the polyelectrolytes of seven-layer PMCs, formed on a CaCO_3_ core, was shown, and the influence of the ionic strength of the solution on the desorption of polyelectrolytes of PMCs was suggested. Therefore, in this work, the desorption of polyelectrolytes of seven-layer ((PAH/PSS)3PAH) PMC_Ps_, PMC_Ca_ and PMC_Mn_ in NaCl solutions with concentrations of 10, 20, 100, 200 (Figure 2) and 1000, 2000 mmol/l (Figure 3) was studied.

As shown in Figure 2, increasing the NaCl concentration from 0 to 200 mM does not result in significant differences in the fluorescence intensity of the supernatant for PMC_Ca_, indicating a similar polyelectrolyte desorption under these conditions. In the case of PMC_Ps_ and PMC_Mn_, an increase in fluorescence intensity is observed at 100 and 200 mM NaCl, suggesting enhanced polyelectrolyte desorption with increasing ionic strength. However, differences in desorption behavior between PMC_Ps_ and PMC_Mn_ are evident. For PMC_Ps_, the fluorescence intensity increases gradually at 100 and 200 mM NaCl, whereas for PMC_Mn_, the fluorescence intensity remains constant at these NaCl concentrations. These findings support our hypothesis that the desorption of polyelectrolyte layers varies depending on the type of core used. The differences in polyelectrolyte desorption can be attributed to the fact that PMC_Ca_ possess a complex internal structure composed of a polyelectrolyte complex [36]. Consequently, this type of capsule has a larger interaction area with the incubation medium compared to PMC_Ps_ and PMC_Mn_, which exhibit a well-defined shell and lack a polyelectrolyte complex within the internal cavity [26,66,73].

The effect of higher concentrations of sodium chloride (1M and 2M), as well as 1M so-dium sulfate, whose ionic strength is equivalent to 3M sodium chloride, was further investigated. The results are shown in Figure 3.

Figure 3 shows that the fluorescence intensity of the supernatant after 48 h of incubation increases with an increasing sodium chloride concentration. Such an effect is consistent with the expectation of the effect of high ionic strength on the desorption of polyelectrolyte multilayers in both polyelectrolyte films [32] and polyelectrolyte microcapsules [62]. Such an effect is attributed to the occurrence of polyelectrolyte charge shielding due to the high salt concentration of all types of PMCs.

However, in the case of PMC_Ps_, as shown in Figure 3B, the dissociation of the polyelectrolyte reaches its maximum value at a lower NaCl concentration (1000 mM), unlike PMC_Ca_ and PMC_Mn_, where an increase in dissociation is observed even at 2000 mM.

In the next stage, we investigated the effect of sodium sulfate on the desorption of PMC_Ps_, PMC_Ca_, and PMC_Mn_. Previous studies [64,74] have demonstrated that polyelectrolyte microcapsules respond differently to the ionic composition of the medium, necessitating an examination of the influence of divalent salts on polyelectrolyte desorption. To this end, polyelectrolyte microcapsules were incubated in salt solutions with concentrations ranging from 5 to 50 mM. The results are presented in Figure 4.

As seen in the figure, the fluorescence intensity in PMC_Mn_, PMC_Ps_ and PMC_Ca_ samples increases proportionally with the rising concentration of Na_2_SO_4_. The increase in fluorescence intensity at such a low ionic strength (equivalent to 15–30 mM NaCl) was not observed in these samples during incubation in NaCl, suggesting that the mechanism of polyelectrolyte desorption differs and may be associated with the specific interaction of sodium sulfate with the amino groups of PAH.

The effect of higher concentrations of sodium sulfate (100–1000 mM) and 1000 mM sodium chloride, whose ionic strength corresponds to 333 mM sodium sulfate, was also investigated. The results are shown in Figure 5.

As shown in Figure 5, the fluorescence intensity from 100 to 1000 mM Na_2_SO_4_ does not differ significantly across all types of PMC samples. However, the desorption level of PMC_Mn_ at 50 mM Na_2_SO_4_ already reaches its maximum value and does not significantly differ from the desorption observed at concentrations ranging from 100 to 1000 mM. This result further supports the hypothesis that the type of core influences the desorption behavior of PMC_Mn_, PMC_Ps_ and PMC_Ca_.

Given that the lowest fluorescence intensity is observed at 1 M NaCl, even when compared to 100 mM Na_2_SO_4_ (equivalent to 300 mM NaCl), it can be inferred that the shielding effect occurs only up to 1 M NaCl. Beyond this point, the further increase in fluorescence intensity is likely associated with a specific desorption mechanism of the polyelectrolyte, linked to the nature of Na_2_SO_4_.

In the literature to date, no similar mechanism has been observed where polyelectrolyte desorption increased at low concentrations of sodium sulfate (less than 100 mM), and the maximum desorption was reached at a concentration of just 100 mM, exceeding polyelectrolyte desorption in the presence of 1M NaCl. Earlier studies posited that, as in the case of polyelectrolyte microcapsules formed on CaCO_3_ cores, both protein-filled [61] and protein-free [62], as well as in other studies, the prevailing effect on the dissociation of polyelectrolyte microcapsules was ionic strength, not the nature of the polyelectrolyte itself. However, this study demonstrates the contrary, suggesting that the type of salt impacts the desorption of polyelectrolytes from PMCs. The differing effects of salts on the properties of polyelectrolyte microcapsules have been noted in other studies. In the work of Heuvingh et al. [26], the authors showed that the salt anion, rather than the cation, has a greater impact on the size change of the PMC, while the SO_4_^2−^ ion has a much smaller effect (an11% reduction) than Cl^−^. They suggested that this is consistent with the position of anions in the Hofmeister series, which classifies ions according to their lyotropic properties. It can be hypothesized that the differing impact of sodium sulfate and sodium chloride salts on the desorption of polyelectrolytes from the shell of microcapsules could be associated with the nature of the interaction between Hofmeister-series salts and the polyelectrolytes of the PMC shell. We report the novel finding that the polyelectrolyte desorption profile from microcapsules is determined by the nature of the sacrificial core template (MnCO_3_, CaCO_3_, PS). This core-specific influence persists despite the core’s complete removal in the last stage of microcapsule preparation, representing a key insight from our research. The key findings of this study are summarized in Table 1.

As shown in Table 1, the desorption of polyelectrolyte microcapsules varies depending on the type of core used during their formation. The differences in polyelectrolyte desorption may be attributed to a combination of factors, including the chemical properties of the cores, their surface structure, the core removal process and its byproducts.

When MnCO_3_ and CaCO_3_ are used as cores for PMCs, there are common features that may influence the desorption of the polyelectrolyte layer, particularly during the dissolution process:1.The release of Ca^2+^/Mn^2+^ and CO_3_^2−^ ions, which can affect local pH and ionic strength [75]. This may alter the electrostatic interactions between polyelectrolytes, thereby influencing the desorption of PAH.2.The released Ca^2+^/Mn^2+^ ions screen charges, reducing repulsion and promoting the contraction of PAH chains (transition from helix to globule) [76,77].3.The rapid release of Ca^2+^/Mn^2+^ during EDTA dissolution creates transient osmotic gradients, which can cause the swelling or rupture of thinner polyelectrolyte shells [36,78].

However, differences also exist. To examine the differences, it is first necessary to evaluate the morphology of the cores on which the PMC shell forms (Figure 6).

In contrast to PS particles, calcium carbonate (CaCO_3_) and manganese carbonate (MnCO_3_) particles deviate from perfect sphericity and surface smoothness, exhibiting a more complex and rougher topography that may influence the resultant polyelectrolyte microcapsule’s morphology and properties. The primary distinction of CaCO_3_ cores from other types lies in their complex internal channel-like organization [36], which results in PMCs lacking a distinct shell and having an internal space filled with an interpolyelectrolyte complex. In contrast, MnCO_3_ cores do not exhibit a channel-like structure but possess a rougher and more complex surface topography compared to polystyrene cores [65]. The absence of a channel-like structure in these types of capsules leads to the formation of polyelectrolyte microcapsules with a distinct shell that lacks an internal polyelectrolyte complex [81,82]. These differences may influence the interaction area between the PMC shell and the incubation medium, as well as microfluidic parameters.

A key distinguishing feature of polystyrene (PS) cores compared to other types is the core removal process. The dissolution of polystyrene particles requires the use of organic solvents, such as dimethylformamide (DMF). Literature reports demonstrate that organic solvents can significantly influence polyelectrolyte microcapsules (PMCs), altering their porosity, size, shell density and other properties [83,84,85]. Consequently, the treatment of PMCs with such solvents may have a substantial impact on desorption behavior. Specifically, the work of Christophe De’jugnatand Gleb B. Sukhorukov [66] has already demonstrated the influence of tetrahydrofuran, a solvent used to remove the templates of PMCs formed on PS cores, on the stability of PMCs formed on MnCO_3_. After 15 h of incubation in this solvent, the shell stability of PMCs formed on MnCO_3_ was significantly increased in an alkaline medium.

Furthermore, desorption of the polyelectrolyte shell may be affected by the intermixing of polyelectrolyte layers that takes place following core removal—an effect we first described in prior work for CaCO_3_-templated microcapsules, showing a good correlation with PMC zeta potential (Table 2).

As seen in Table 2, the shell’s zeta potential is determined by the outer layer only before core dissolution. After removal, the zeta potential becomes fixed, irrespective of the outer layer. The polyelectrolyte layer intermixing effect prevents us from obtaining data that would enhance our understanding of the influence of these parameters on the desorption of shell polyelectrolytes from PMCs depending on the core type.

Variations in the size of microcapsules formed on different cores (MnCO_3_: 3 ± 1 µm; CaCO_3_: 4.5 ± 1 µm; PS: 5 ± 0.2 µm), and their differential size changes in response to varying salt types and concentrations, represent another potential factor for the observed differences in polyelectrolyte desorption behavior (with literature reports indicating stable PMC dimensions post-core removal [65]). Size differences may alter the interaction surface area between the shell polyelectrolytes and the incubation medium. We have also previously shown differing responses to ionic strength [65]: PS-templated microcapsules shrink (by 11%, with anion-specific effects), while CaCO_3_-templated capsules retain their original dimensions irrespective of ionic strength.

The observed high sensitivity of PMC shell desorption to ionic strength and the nature of the salt primarily necessitates careful selection of the appropriate capsule type for specific environmental conditions. For instance, in biomedical applications such as targeted drug delivery, the choice of PMC_Ca_ might be preferable for environments with a low ionic strength, while PMC_Ps_ or PMC_Mn_ could be more suitable for conditions involving a higher ionic strength or specific salt compositions. This sensitivity can also be leveraged in the development of smart materials, where a specific type of capsule could be designed to either degrade or remain stable under predetermined ionic strength conditions. Thus, the ability to control PMCs behaviour based on core material opens new possibilities for designing advanced functional materials with an accurate responsiveness to environmental stimuli.

## 4. Conclusions

Within the framework of this study, the desorption of shell polyelectrolytes from polyelectrolyte microcapsules (PMCs) formed on MnCO_3_ and polystyrene cores was investigated for the first time, and a comparative analysis was conducted with previously obtained data on the desorption of PMCs formed on CaCO_3_ cores. The results revealed that the desorption of polyelectrolytes from PMCs varies depending on the type of core used. Low ionic strength solutions (up to 200 mM NaCl) have a negligible effect on the desorption of PMC_Ca_. In contrast, for PMC_Ps_ and PMC_Mn_, an increase in desorption is observed at 100 and 200 mM NaCl. Increasing the ionic strength to 1000 mM and 2000 mM leads to a gradual increase in polyelectrolyte desorption for PMC_Ca_ and PMC_Mn_, while for PMC_Ps_, maximum desorption is already achieved at 1000 mM. Additionally, an increase in desorption was observed during incubation in various concentrations of sodium sulfate (5–50 mM), with desorption levels differing only slightly across all types of PMCs. However, for PMC_Mn_, maximum desorption occurs at 50 mM sodium sulfate, whereas for other capsule types, maximum desorption is observed at 100 mM. Given that desorption at 1 M NaCl is lower than at 100 mM Na_2_SO_4_ (I = 300 mM), it can be hypothesized that desorption is not solely due to electrostatic shielding but also involves a specific mechanism characteristic of Na_2_SO_4_. This study establishes that the sacrificial core template (CaCO_3_, MnCO_3_ or PS) fundamentally defines the physicochemical behavior of polyelectrolyte microcapsules (PMCs) and the effect on the desorption processes of polyelectrolyte, even after core removal.

The aforementioned results demonstrate that the desorption behaviors of PMC_Ca_, PMC_Ps_ and PMC_Mn_ differ and exhibit distinct patterns depending on the concentration of sodium chloride and sodium sulfate in the medium. These findings will enable researchers in this field to accurately interpret the results obtained from PMCs formed on different types of cores and to select the appropriate capsule type for specific applications in studies focused on the development of PMC-based targeted drug delivery systems, diagnostic tools and smart materials.

## Figures and Tables

**Figure 1 polymers-17-01706-f001:**
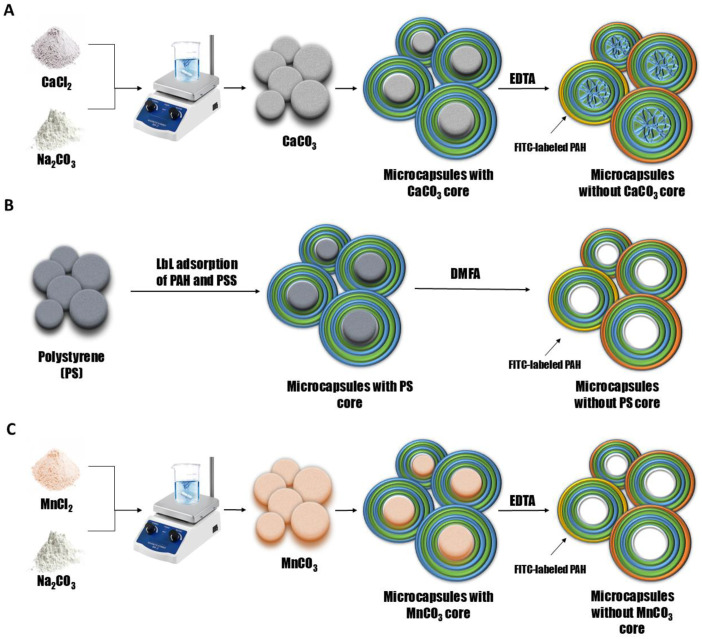
Scheme of preparation of polyelectrolyte microcapsules: PMC, formed on CaCO_3_ core (**A**) (adapted from [70]); PMC, formed on PS core (**B**); PMC, formed on MnCO_3_ core (**C**).

**Figure 2 polymers-17-01706-f002:**
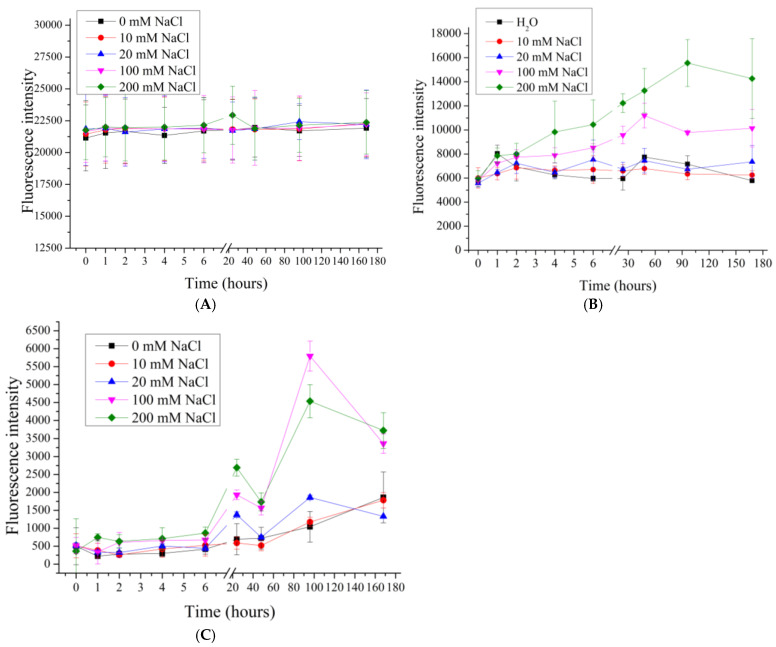
Fluorescence intensity of supernatant from PMC samples in NaCl solution (0 to 200 mM). PMC, formed on CaCO_3_ core (**A**); PMC, formed on PS core (**B**); PMC, formed on MnCO_3_ core (**C**).

**Figure 3 polymers-17-01706-f003:**
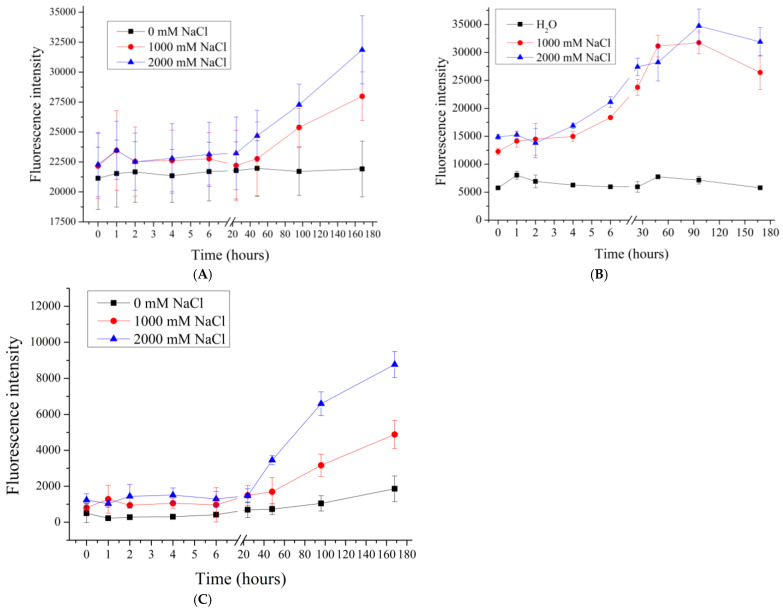
Fluorescence intensity of supernatant from PMC samples in NaCl solution (1000 to 2000 mM). PMC, formed on CaCO_3_ core (**A**); PMC, formed on PS core (**B**); PMC, formed on MnCO_3_ core (**C**).

**Figure 4 polymers-17-01706-f004:**
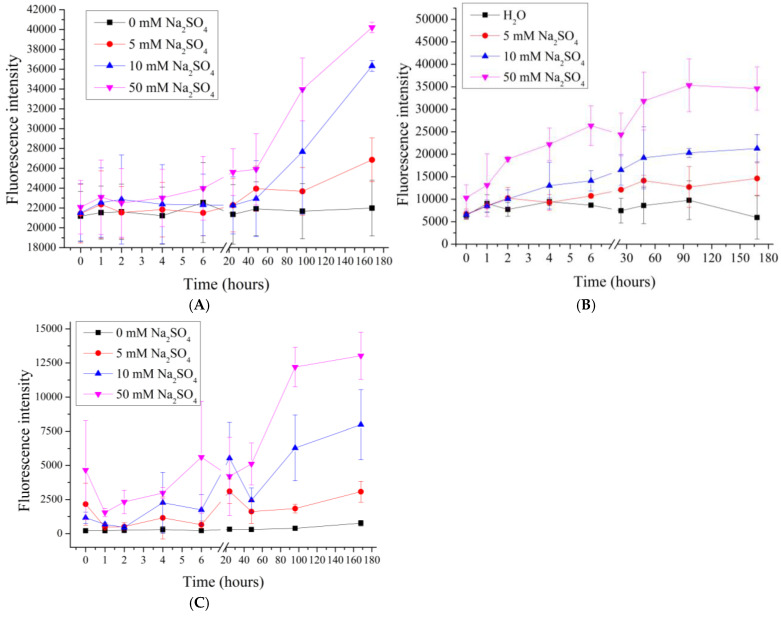
Fluorescence intensity of supernatant from PMC samples in Na_2_SO_4_ solution (0 to 50 mM). PMC, formed on CaCO_3_ core (**A**); PMC, formed on PS core (**B**); PMC, formed on MnCO_3_ core (**C**).

**Figure 5 polymers-17-01706-f005:**
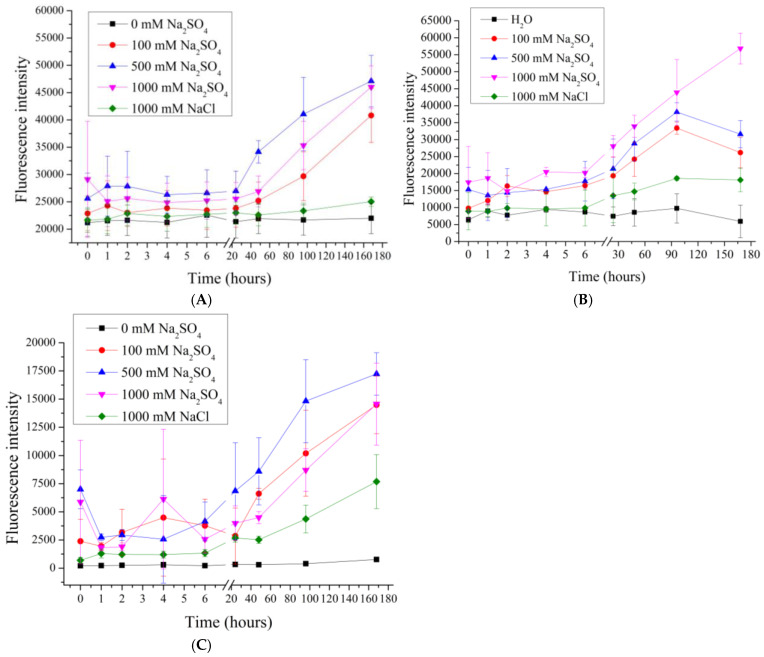
Fluorescence intensity of supernatant from PMC samples in Na_2_SO_4_ solution (100 to 1000 mM). PMC, formed on CaCO_3_ core (**A**); PMC, formed on PS core (**B**); PMC, formed on MnCO_3_ core (**C**).

**Figure 6 polymers-17-01706-f006:**
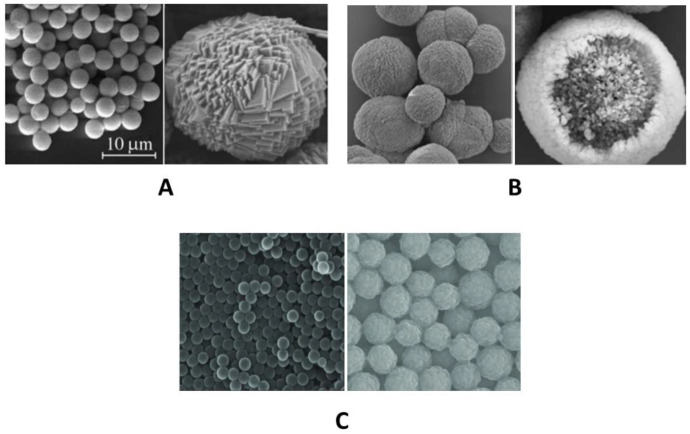
SEM images of (**A**) MnCO_3_ [79]; (**B**) CaCO_3_ [79]; and (**C**) polystyrene [80] particles. Adapted from [65,79,80].

**Table 1 polymers-17-01706-t001:** Influence of core composition on polyelectrolyte microcapsule desorption in NaCl and Na_2_SO_4_. Max.des.—maximal desorption.

	NaCl		Na_2_SO_4_	
Type of PMC	Up to 200 mM	1000 to 2000 mM	Up to 50 mM	100 to 1000 mM
PMC_Ca_	Minimal desorption	Gradual increase	Gradual increase	Max. des. at 100 mM
PMC_PS_	Gradual increase	Max. des. at 1000 mM	Gradual increase	Max. des. at 100 mM
PMC_Mn_	Gradual increase	Gradual increase	Max. des. at 50 mM	Max. des. at 50 mM

**Table 2 polymers-17-01706-t002:** Zeta potential of microcapsules formed on CaCO_3_ before and after dissolution of core. Adapted from [86].

	PMC Before Core Dissolution	PMC After Core Dissolution
CaCO_3_ [86]		
[PAH/PSS]_3_	−20 ± 3.4 mV	+25 ± 3.1 mV
[PAH/PSS]_3_PAH	+24 ± 2.7 mV	+23 ± 3.4 mV

## Data Availability

Data is contained within the article.

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
