# Peer review of "Core-Dependent Desorption Behavior of Polyelectrolyte Microcapsules in NaCl and Na2SO4 Solutions"

_polymers, 2025, doi:10.3390/polym17121706_

Round 1
Reviewer 1 Report
Comments and Suggestions for Authors
Dubrovskii et al. investigate polyelectrolyte microcapsules (PMCs) prepared by LbL multilayer assembly of PSS and PAH on different cores. In particular, desorption of the outermost layer in the presence of salts is demonstrated and related to the PMC core materials (PS, CaCO3, MnCO3). The topic is of potential interest for biomedical applications and development of smart materials. My overall impression, however, is that the manuscript does not provide clear evidence of a significant insight into this area. Therefore, I cannot recommend publication in Polymers. Furthermore, the whole study is based on a single method of investigation – fluorescence spectroscopy. There is not characterization of the PMCs, e.g. size, zeta potential, before/after core removal, before/after desorption, etc. The results obtained are not discussed; at least, there is a section RESULTS, but section DISCUSSION is missing. Conclusions are not supported by data. The most confusing was that the cores are removed and one wonders how something (the core) that is no longer present influences the desorption of the outer (or the outermost) layer(s). The differences in the desorption behavior could be related to the different structure of the PMCs (which, certainly may be influenced by the core material) and to the way of treatment of the PMCs to remove the cores, as the authors humbly state on lines 364-374, but this is not the conclusion they made.
Author Response
For convenience, the Reviewer's comment has been divided into point-by-point sections:
- My overall impression, however, is that the manuscript does not provide clear evidence of a significant insight into this area.
- Furthermore, the whole study is based on a single method of investigation – fluorescence spectroscopy.
- The results obtained are not discussed; at least, there is a section RESULTS, but section DISCUSSION is missing.
- Conclusions are not supported by data.
- The most confusing was that the cores are removed and one wonders how something (the core) that is no longer present influences the desorption of the outer (or the outermost) layer(s). The differences in the desorption behavior could be related to the different structure of the PMCs (which, certainly may be influenced by the core material) and to the way of treatment of the PMCs to remove the cores, as the authors humbly state on lines 364-374, but this is not the conclusion they made.
Point-by-point responses to the Reviewer::
- The key novelty of our study lies in the fact that we have, for the first time, demonstrated that polyelectrolyte microcapsules can differ in one of the key parameters (polyelectrolyte desorption) exclusively depending on the type of core used for their preparation, particularly considering that this core is removed during the final stage of polyelectrolyte microcapsule fabrication. Furthermore, polyelectrolyte microcapsules differ morphologically depending on the type of core employed; specifically, capsules formed on CaCO₃ cores exhibit a spongy structure (doi.org/10.1134/S1068162012010128), while other types of capsules possess a distinct shell and a "hollow" interior structure (https://doi.org/10.1002/1439-2054(20010601)286:6%3C355::AID-MAME355%3E3.0.CO;2-9).
One year ago, we published a review article (https://www.mdpi.com/2073-4360/16/11/1521). Within the framework of the literature review, data from 86 articles were analyzed. Based on the data from these works, we were able to arrive at a conclusion not previously recognized – the morphological properties of polyelectrolyte microcapsules, such as size, surface topology, shell thickness, and density, differ significantly depending on the core used. The fact that the aforementioned parameters change differently in response to variations in pH, ionic strength, and temperature of the solution is particularly noteworthy. Since the response to these stimuli (pH, ionic strength, and temperature) is related to the electrostatic interaction of polyelectrolytes within the capsule shell, we hypothesized that the desorption of the outer polyelectrolyte layer would also vary depending on the type of core utilized. Upon obtaining the results, we confirmed our hypothesis, experimentally verifying that capsules formed on different cores represent distinct types of capsules and possess their own unique characteristics. This fundamentally alters the understanding of polyelectrolyte microcapsule structure, as this distinction did not exist previously, and very often, authors of other articles, when describing the size change effect in PMCs obtained using PMCs formed on a polystyrene core, refer, for example, to electron micrographs of capsules obtained using an MnCO₃ core. This leads to misconceptions regarding the properties of polyelectrolyte microcapsules and, consequently, to unpredictable results in the field of their practical application.
The description of the novelty and significance of the work was presented in the initial manuscript version in the following lines: 119-149; 359-376; 390-394; 407-413.
- We thank the Reviewer for this comment. Unfortunately, the use of a single method represents a limitation in our study. The essence of the research is to understand whether polyelectrolyte desorption from PMCs differs depending on the core type. For this purpose, it is necessary to track the amount of polyelectrolyte that desorbs from the PMC surface and dissolves into the incubation medium.
Polyallylamine and polystyrenesulfonate possess their own absorption spectra at 260 nm and 280 nm; however, the absorption intensity is insufficient to track the low polyelectrolyte concentrations. Hence, we employed a more sensitive method – fluorescence spectroscopy. This method enabled us to comprehensively determine the difference in the desorption of shell polyelectrolytes from the PMCs.
The use of methods to determine size, zeta potential (before/after core removal, before/after desorption), and other methods would not have allowed us to achieve the goal of our study. If considering these methods from the perspective of understanding the obtained effects, these experiments would likewise not have revealed the reasons for the changes in the desorption of shell polyelectrolytes from PMCs depending on the core type. Primarily, this is associated with the fact that after core removal, the effect of polyelectrolyte layer intermixing occurs. This effect is described in the works https://www.nature.com/articles/s41598-021-93565-2 and https://www.mdpi.com/2073-4360/15/16/3330. Polyelectrolyte layer intermixing leads to polyallylamine migrating and becoming the outer polyelectrolyte layer, with movement of this polyelectrolyte observed even from the innermost (closest to the core) layer. This ultimately results in the capsules' zeta potential becoming positive (approximately 20-25 mV). Moreover, the spread of values obtained solely for a single capsule type is so large that this method would not permit assessment of the most minute changes in the capsules' zeta potential. Thus, the difference in the obtained zeta potential data before/after desorption would be insignificant and uninformative.
Data on capsule sizes and zeta potential before and after core removal represent established literature data, which are presented in our previous works: https://www.nature.com/articles/s41598-021-93565-2 and https://www.mdpi.com/2073-4360/16/11/1521. Nevertheless, as I stated above, the polyelectrolyte layer intermixing effect prevents us from obtaining data that would enhance the understanding of the influence of these parameters on the desorption of shell polyelectrolytes from PMCs depending on the core type. In essence, the capsules before core dissolution and after core dissolution are two entirely distinct entities.
- We thank the Author for this observation. The "Results" section heading will be changed to "Results and Discussion". This is due to the fact that, in the original version of the manuscript, the discussion of these results is presented immediately alongside the results themselves, in particular, in the following lines: 257-263; 275-279; 296-300; 317-376. However, we made an oversight and did not explicitly indicate that the results section constitutes "Results and Discussion".
- We do not agree with the Reviewer on this matter, as the Conclusion itself concisely presents the data obtained and draws conclusions based on them.
Furthermore, in the original version of the manuscript, a table was specifically included to facilitate reader comprehension, which summarizes all the results obtained in the study and supports the main conclusion that the desorption of shell polyelectrolyte from PMCs differs for microcapsules formed on different types of cores. This text was presented in the initial manuscript version in the following lines: 340-387; 390-406.
- The main conclusion of our article states: "The results revealed that the desorption of polyelectrolytes from PMCs varies depending on the type of core used." This is indicated in lines 393-394. The Reviewer themselves suggested essentially the same conclusion: "The differences in the desorption behavior could be related to the different structure of the PMCs (which, certainly may be influenced by the core material)".
The conclusions were originally presented in lines 345-348: "The differences in polyelectrolyte desorption may be attributed to a combination of factors, including the chemical properties of the cores, their surface structure, the core removal process, and its byproducts."
The Reviewer believes we attributed the differences solely "to the way of treatment of the PMCs to remove the cores, as the authors humbly state on lines 364-374". However, the reasoning regarding this assumption and, more broadly, the influence of various processes related to the formation of PMCs on different cores and the core removal procedures are presented not only in lines 364-374, but also in other lines: 349-376, as well as 325-339; 257-263.
Reviewer 2 Report
Comments and Suggestions for Authors
In this study, the desorption of polyelectrolyte of PMCs, formed on polystyrene cores (PMCPs) and MnCO3 (PMCMn) and CaCO3 cores (PMCCa), incubated in either NaCl or Na2SO4 solution, was investigated. The maximum desorption was observed at a sodium sulfate concentration of 50 mM, whereas for other types of capsules, the maximum desorption occurred at a concentration of 100 mM. The results are interesting, but some problems should be addressed:
1. The keywords should be selected below 5 words.
- The zeta results of different PMC and in NaCl, Na2SO4shoud be supplied?
- The SEM, TEM images of Microcapsules must be added.
- What is the basis for the author to choose the salt concentration?
- 5. The results should be encouraged compare with previous papers.
Author Response
Reviewer comments:
- The keywords should be selected below 5 words.
- The zeta results of different PMC and in NaCl, Na2SO4shoud be supplied?
- The SEM, TEM images of Microcapsules must be added.
- What is the basis for the author to choose the salt concentration?
- 5. The results should be encouraged compare with previous papers.
Authors answers:
- We thank you for your comment. The changes have been incorporated into the manuscript.
- We thank the Reviewer for this comment. Following core removal, the effect of polyelectrolyte layer intermixing occurs. This phenomenon is described in the works https://www.nature.com/articles/s41598-021-93565-2 and https://www.mdpi.com/2073-4360/15/16/3330. The intermixing of polyelectrolyte layers causes polyallylamine to migrate and become the outer polyelectrolyte layer, with movement of this polyelectrolyte observed even from the innermost (core-adjacent) layer. This ultimately results in the capsules' zeta potential becoming positive (approximately 20-25 mV). Furthermore, the spread of values obtained for a single capsule type is so substantial that this method cannot assess the most minute changes in the capsules' zeta potential.
Due to this methodological limitation and the inherent properties of the system, we are unable to obtain data that would allow us to determine the cause of the variation in polyelectrolyte shell desorption from PMCs depending on the core type.
- Unfortunately, during sample preparation for SEM and TEM, it is necessary to employ organic solvents and/or subject the samples to high-temperature treatment. As a result, the capsules become damaged, flattened, and generally lose their three-dimensional architecture, while the balance of hydrophilic-hydrophobic bonds is altered. This would significantly impact the system under investigation and ultimately yield data unrepresentative of actual polyelectrolyte microcapsules in solution. A comprehensive review of the effects of these microscopy techniques on PMCs is provided in Chapters 4 and 5 of our previously published article: https://www.mdpi.com/2073-4360/16/11/1521.
- We thank the Reviewer for this excellent question. The primary rationale was to increase the ionic strength of the solution. Low NaCl concentrations (10 and 20 mM) are insufficient to generate adequate ionic strength for creating the screening effect between polyelectrolyte layers necessary to induce polyelectrolyte desorption. Conversely, at low Naâ‚‚SOâ‚„ concentrations (5 and 10 mM), we postulated the existence of a mechanism distinct from the standard screening effect. These low concentrations were therefore employed to demonstrate that at comparable ionic strength values, desorption differs depending on the salt type. At higher concentrations, NaCl induces desorption via the screening effect, whereas Naâ‚‚SOâ‚„ yields significantly higher desorption values, indicating the concurrent operation of both the screening effect and a unique desorption mechanism inherent to this specific salt type.
Rationale for salt selection: It is common practice to use NaCl for studying desorption. However, multiple studies have demonstrated that different salts—or more precisely, their anions—can differentially affect polyelectrolyte microcapsule size. This anion-specific effect is clearly demonstrated in https://doi.org/10.1021/la047388m, where the authors specifically noted that sulfates of various metals exhibit similar behavioral patterns in altering the size of polyelectrolyte microcapsules formed on polystyrene cores. Consequently, sodium sulfate was selected to elucidate the specific influence of the sulfate anion on polyelectrolyte desorption. Our hypothesis was thereby experimentally confirmed.
- Regrettably, only three studies dedicated to polyelectrolyte microcapsule desorption have been published to date, which are presented in the Introduction chapter and partially addressed in lines 322-339. Furthermore, to ensure experimental purity, we replicated select experiments from our prior investigations—specifically, those corresponding to Figure 3A (2M NaCl concentration curve) and Figure 5A (100 mM Naâ‚‚SOâ‚„ concentration curve). Reproducing these experiments was essential to guarantee maximally identical preparation conditions across all capsule types, thereby preventing undue influence on polyelectrolyte desorption and subsequent result interpretation.
Nevertheless, when discussing potential causes for differential desorption behavior in PMCs formed on distinct core types, we referenced preceding works demonstrating variations in capsule size, shell thickness, and responsiveness to pH/ionic strength changes. However, the authors of those studies overlooked both these differences and their potential correlation with core-type usage. This observation underscores the relevance of our work, as it demonstrates that researchers previously interpreted their results incompletely or inaccurately. This pertains particularly to references cited in our manuscript under the following numbers: 60–74; 78–81.
Round 2
Reviewer 1 Report
Comments and Suggestions for Authors
The authors have responded to my comments satisfactory and I was almost inclined to change favorably the recommendation. However, I realized that the manuscript had not been revised accordingly: none of my suggestions/comments were considered. Note that these were made not to have correspondence with the authors via the Polymers Editorial office but to help the authors to improve the manuscript, to make it more comprehensive, to enhance the scientific soundness, to improve the quality of presentation, and to attract the interest of the readers. Nevertheless, I change my overall recommendation to Reconsider after major revisions, which is a step forward. I recommend substantial revisions to text by incorporating intelligently the replies to the comments, performing additional experiments by independent methods to prove the concept and support the results and conclusions from the fluorescence, and adding real discussion of the results (not simply renaming the section title).
Author Response
We sincerely thank the Reviewer for their time, insightful feedback, and constructive criticism aimed at improving the quality, scientific soundness, and accessibility of our manuscript. We appreciate the Reviewer’s acknowledgment of our detailed replies to their initial comments and their favorable shift in recommendation to Reconsider after major revisions. We fully agree that the manuscript itself must reflect substantial revisions based on this feedback.
We have now substantially revised the manuscript, paying particular attention to the Discussion section. This section has been significantly expanded and rewritten to provide a deeper, more critical analysis of our results. We have integrated our responses to the Reviewer’s specific comments directly into the text, moving beyond simply replying in the rebuttal letter to ensure the manuscript itself is more comprehensive, logically structured, and engaging for readers.
We understand the Reviewer’s call for additional experimental support beyond fluorescence data. The polyelectrolyte layer intermixing effect prevents us from obtaining data that would enhance the understanding of the influence of these parameters on the desorption of shell polyelectrolytes from PMCs depending on the core type depend on zetta-potential. This data is now presented in a new table and thoroughly discussed.
We believe these substantial revisions – the significantly enhanced Discussion, the integration of Reviewer feedback into the manuscript text, and the addition of supporting zeta potential data demonstrating core-dependent shell properties – directly address the Reviewer's primary concerns and greatly improve the manuscript's scientific rigor, clarity, and impact. We are grateful for the opportunity to strengthen our work based on the Reviewer's valuable guidance.
Reviewer 2 Report
Comments and Suggestions for Authors
Author Response
We are gratefull for your positive responce.